# Finger fractures: Epidemiology and treatment based on 21341 fractures from the Swedish Fracture register

**Henrik Alfort** [1,2]*, **Johanna Von Kieseritzky** [1,2], **Maria Wilcke** [1,2]

1 Department of Clinical Science and Education, Karolinska Institutet, Södersjukhuset, Stockholm, Sweden,
2 Department of Hand Surgery, Södersjukhuset, Stockholm, Sweden

* henrik.alfort@ki.se

**Data Availability Statement:** All relevant data are within the paper and its Supporting Information files.

## Abstract

### Background

There is a lack of detailed epidemiological studies of finger fractures, the most common fracture of the upper extremity.

### Methods

Based on data of 21 341 finger fractures in the Swedish Fracture register, a national quality registry that collects data on all fractures, this study describes anatomical distribution, cause, treatment, age distribution, and result in terms of patient related outcome measures (PROMs).

### Results

The most common finger fracture was of the base of the 5th finger, followed by the distal phalanx in the 4th finger. Open fractures were most common in the distal phalanges, especially in the 3rd finger. Intraarticular fractures were most frequent in the middle phalanges. Fall accidents was the most common cause of a fracture. The mean age at injury was 40 years (38 for men, 43 for women). 86% of finger fractures in adults were treated non-operatively. Men were more frequently operated than women. Finger fractures did not affect hand function or quality of life and there were no relevant differences in PROMs between fracture type, treatment, or sex.

### Conclusion

This study presents detailed information about the various types of finger fractures which can be used as point of reference in clinical work and for future studies.

**Funding:** This work was supported by grants from the Regional Agreement on Medical Training and Clinical Research (ALF) between the Stockholm County Council and Karolinska Institute, (FoUI-960047, JVK, https://ki.se/en/about/national-and-regional-alf-agreements). The funders had no role in study design, data collection and analysis, decision to publish, or preparation of the manuscript.

**Competing interests:** The authors have declared that no competing interests exist.

# Introduction

## Background

Finger fractures are the most common fractures in the upper extremity [1–3]. They affect patients of all ages and may cause impaired hand function and disability due to pain, malunion and stiffness [4]. Even uncomplicated cases may lead to inability to work for many months after the injury [5] and there is a risk for prolonged use of opiates after surgical treatment of hand fractures [6]. Most finger factures are treated non-operatively with a plaster cast, but some fractures are unstable and require surgical fixation with pins, screws, or plates [7]. The treatment of finger fractures has been reported to vary due to sex, age, social status as well as fracture type [8,9].

The epidemiology of finger fractures is yet described only in limited populations or as sports related injuries, and without regard to detailed information about fracture location or type [2,10–13].

A national quality registry is a population-based collection of individual clinical data on a specific diagnosis, treatment, or outcome. Data from these registries can be used to monitor quality of health care and results but can also be used in research. The Swedish Fracture Register (SFR) collects population-based data on fractures of all types since 2011 [14]. Today, 54 Swedish orthopedic and trauma units report to SFR. Patients seeking care for a fracture are informed about the registry and can choose to not participate, no signed consent is needed according to Swedish legislation. Patients can at any time have their data erased from the registry [15]. The fractures are classified according to the International Statistical Classification of Diseases and Related Health Problems 10th Revision (ICD-10), Arbeitsgemeinschaft für Osteosynthesefragen (AO) and Orthopaedic Trauma Association (OTA) [16–18] by the treating physician at each participating center based on the available radiological information (i.e., plain radiographs). A computer tomography scan or magnetic resonance imaging may be obtained if the attending physician consider it needed. Primary treatment and reoperations are recorded. Patient reported outcome measures (PROMs) in form of the Short Musculoskeletal Function Assessment (SMFA) [19] and quality of life (EQ-5D) [20] are assessed at baseline and one year after the fracture.

The aim of this study was to describe anatomical distribution, treatment, and the incidence of finger fractures based on data from the SFR and to assess possible differences in treatment and result in terms of PROMs according to fracture type, treatment, and sex. Epidemiologic research can identify risk factors, groups at risk and describe current treatment for specific a condition. This knowledge can enable better allocation of resources and more correct implementation of evidence-based treatments [21]. The SFR data represents the majority of the centers treating fractures in Sweden and can provide a broad and accurate epidemiological picture of finger fractures.

# Methods

## Study protocol

This is a cohort study based on data for all finger fractures registered in the SFR for the years 2012–2019. The STROBE guidelines were used for this study [22]. The STROBE checklist is submitted in the supporting information segment. The research was performed according to the Declaration of Helsinki. Ethical approval for this study was obtained from the Swedish Ethical Review Authority (Dnr: 2020–02115). Patient inclusion in a national quality registry such as the SFR is regulated by Swedish legislation and approved by Swedish Data Inspection Board [15].

### Eligibility criteria

The finger fractures registered in the SFR by treating physicians at all contributing trauma centers were identified in the registry with the ICD-10 code S62.6 (phalangeal fractures). Complete data set is submitted in the supporting information segment.

### Outcome assessment

Primary treatment (non-operative or operative) and secondary procedures (i.e., surgery due to failed conservative or surgical treatment, infection or nonunion) were registered. Patient age at the time of injury, sex, cause, and place of injury were recorded. Children (defined as <15 years) were not registered in the SFR until 2015. Therefore, analysis of age and incidence were only performed on data from 2015–2019. In the analysis of fracture type, -location and treatment; children and adults were analyzed separately due to different injury patterns and mechanisms.

SMFA and EQ-5D are distributed electronically to all patients >15 years, at the time of injury (representing the week before the injury) and after one year. Patients who do not answer the questionnaires electronically, receives them in paper form. The SMFA consists of 46 questions (Likert scale 1–5) about musculoskeletal dysfunction and bother. The index is 0–100 where a higher score indicates more dysfunction and bother. The SMFA score can be divided into six categories and the arm/hand function index (0–100) was used in this study. EQ-5D is a widely used generic measure quality of life where 0 equals dead, and 1 equal full health. EQ-5D had originally three levels (EQ-5D-3L) but later a five-level version has become available (EQ-5D-5L). SFR changed from EQ-5D-3L to EQ-5D-5L in 2018. In this report only the EQ-5D-3L was used and patients that answered the 5L version are not included in the analysis of EQ-5D.

The distribution of different fracture types, their age- and sex distribution, treatment and cause of injury was analyzed. SMFA arm/hand index and EQ-5D index were compared between fractured fingers, phalanx (proximal, middle, or distal), intra- or extraarticular fractures, treatment (non-operative or operative) and sex.

The SFR started in Gothenburg and the surrounding Region Västra Götaland (VG-region) and this region remains the area with the best coverage from start. Therefore, only the data from VG-region was used to calculate the incidence. Population data (annual mid-population) was obtained from the database of statistics Sweden (SCB.se).

### Statistics

Differences in treatment according to sex and different fractures was tested with $Chi^2$ test. Changes in EQ-5D and SFMA arm/hand index between before and one year after injury were analyzed with Wilcoxon signed rank test. Differences in EQ-5D and SFMA arm/hand index between sex, treatment and fracture type were analyzed with Kruskal-Wallis and Mann-Whitney U test. Incidence rates were calculated as the number of fractures divided by the total number of person-years (population at risk) and expressed per $10^4$ person-years (PYR). Analyses were performed with IBM SPSS Statistics version 28.0.0.0 (190). Significance level was set at p = 0.05.

## Results

### Fracture epidemiology

21 341 individual finger fractures were identified in the registry. Fig 1 shows a flowchart of all finger fractures found in the SFR. The predominant cause of injury was a fall (29%) followed by

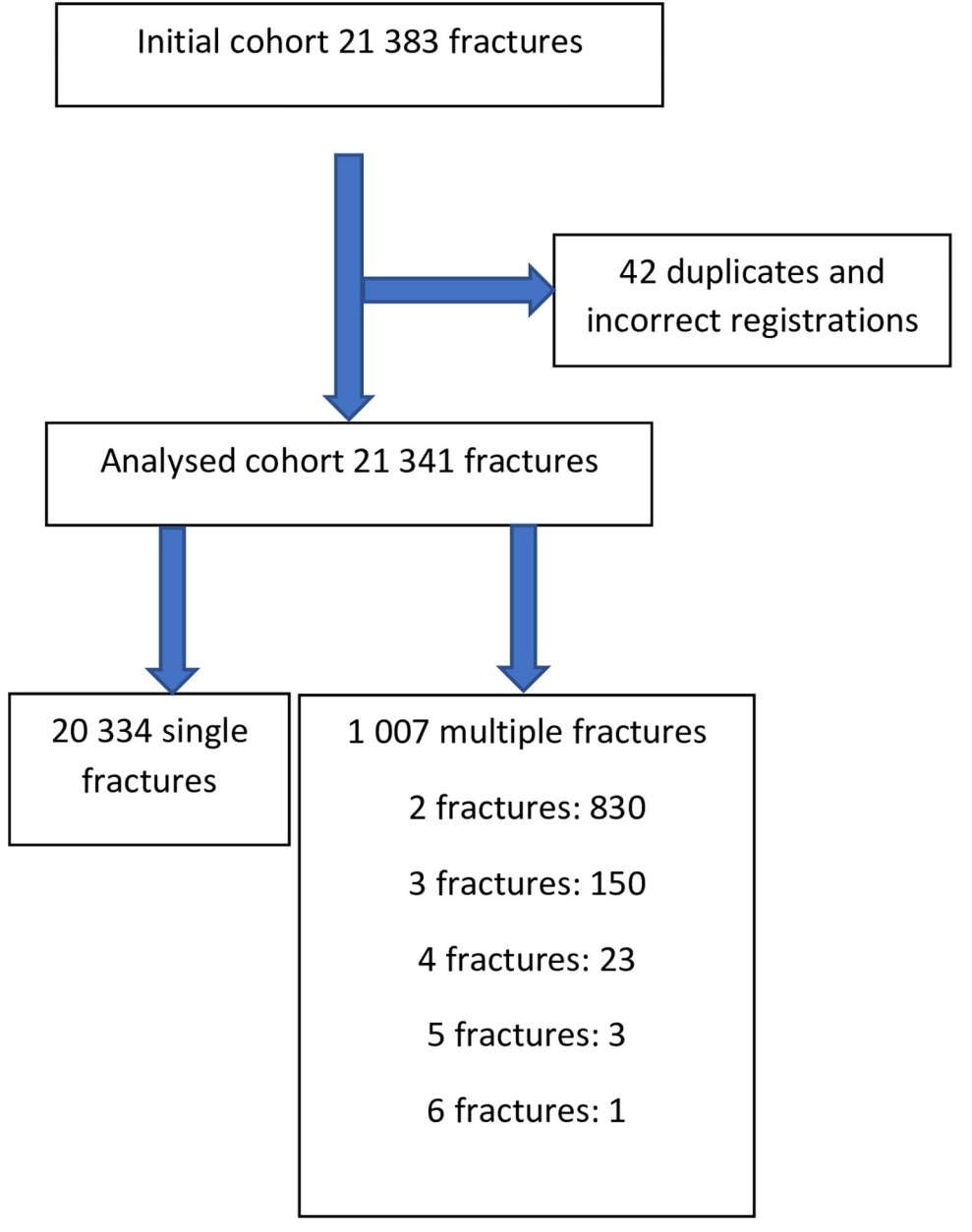

**Fig 1. Flowchart of the registry data analysis.**

crush injury (17%). The mean age at injury was 40 years (range, 0–101 years) with a higher mean age for women than men (43 and 38 years, respectively). From late adolescence to approximately 55 years of age there was a notable discrepancy between the sexes. The age distribution for men and women is presented in Fig 2. Tables 1 and 2 shows the anatomical distribution of the fractures for children and adults respectively. The most common fracture, in adults and children, was a closed fracture of the proximal phalanx in the 5th finger, followed by a closed fracture of the distal phalanx in the 4th finger for adults and a closed fracture of the proximal phalanx of the 4th finger in children. Intraarticular fractures were most frequent in the middle phalanges. 17% of all fractures in adults were open and more common among men (22% compared to 8% in women). Open and intraarticular fractures were less common in children.

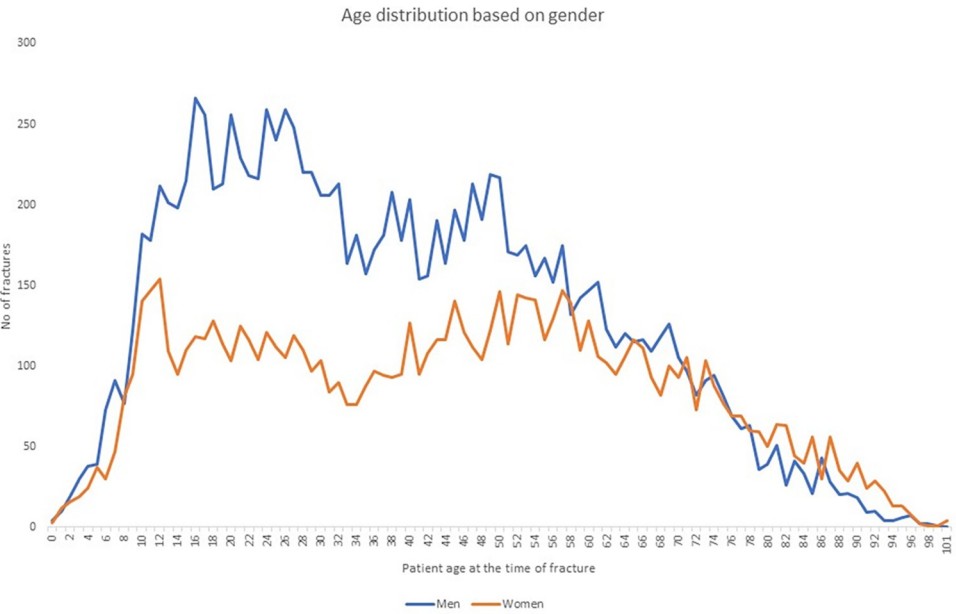

**Fig 2. Age distribution for all finger fractures 2015–2019.** Blue-men, red-women.

## Treatment

Non-operative treatment dominated for all fracture types (86% in adults, 92% in children) Fractures of the distal phalanx in adults were treated non-operatively to a greater extent than fractures in the middle and proximal phalanx (Table 1). Open fractures were operated to a greater extent than closed (34% vs. 10%). In men, 16% of the fractures were operated

**Table 1. The anatomical distribution, presence of intra-articular fracture, mean age, sex ratio, and non-operative vs. operative treatment of finger fractures in adults (15 years and older).**

| Finger | Phalanx | Number | Mean age | Sex (M/F %) | Treatment (non-operative/operative %)* | Open fractures % | Intra-articular % |
|---|---|---|---|---|---|---|---|
| **Dig 2** | **All** | **2899** | **44** | **71/29** | **84/16** | **32** | **32** |
| | Base | 964 | 44 | 73/27 | 80/20 | 17 | 37 |
| | Middle | 579 | 41 | 67/37 | 75/25 | 31 | 59 |
| | Distal | 1356 | 46 | 72/28 | 90/10 | 43 | 17 |
| **Dig 3** | **All** | **3867** | **44** | **62/38** | **87/13** | **23** | **35** |
| | Base | 942 | 48 | 57/43 | 83/17 | 9 | 37 |
| | Middle | 996 | 41 | 53/47 | 86/14 | 16 | 61 |
| | Distal | 1929 | 44 | 69/31 | 90/10 | 34 | 20 |
| **Dig 4** | **All** | **5083** | **46** | **57/43** | **86/14** | **13** | **37** |
| | Base | 1750 | 51 | 49/51 | 82/18 | 5 | 29 |
| | Middle | 1316 | 43 | 50/50 | 85/15 | 8 | 61 |
| | Distal | 2017 | 43 | 69/31 | 91/9 | 24 | 30 |
| **Dig 5** | **All** | **7009** | **47** | **56/44** | **86/14** | **9** | **38** |
| | Base | 4029 | 49 | 51/49 | 86/14 | 4 | 25 |
| | Middle | 1302 | 44 | 55/45 | 85/15 | 9 | 68 |
| | Distal | 1678 | 43 | 70/30 | 87/13 | 20 | 49 |

*Unregistered treatment: 5%.

**Table 2. The anatomical distribution, presence of intra-articular fracture, mean age, sex ratio, and non-operative vs. operative treatment of finger fractures in children (under 15 years).**

| Finger | Phalanx | Number | Mean age | Sex (M/F %) | Treatment (non-operative/ operative %)* | Open fractures % | Intraarticular % |
|---|---|---|---|---|---|---|---|
| **Dig 2** | **All** | **338** | **10** | **60/40** | **94/6** | **9** | **16** |
| | Base | 176 | 10 | 62/38 | 96/4 | 1 | 10 |
| | Middle | 90 | 10 | 59/41 | 95/5 | 7 | 28 |
| | Distal | 72 | 9 | 56/40 | 88/12 | 31 | 14 |
| **Dig 3** | **All** | **428** | **10** | **56/44** | **90/10** | **12** | **19** |
| | Base | 186 | 11 | 56/44 | 94/6 | 1 | 8 |
| | Middle | 95 | 10 | 53/47 | 89/11 | 6 | 39 |
| | Distal | 147 | 8 | 60/40 | 86/14 | 29 | 20 |
| **Dig 4** | **All** | **462** | **10** | **59/41** | **91/9** | **7** | **15** |
| | Base | 235 | 11 | 58/42 | 90/10 | 1 | 10 |
| | Middle | 109 | 11 | 60/40 | 95/5 | 3 | 33 |
| | Distal | 118 | 8 | 61/39 | 90/10 | 22 | 10 |
| **Dig 5** | **All** | **1254** | **10** | **60/40** | **92/8** | **2** | **9** |
| | Base | 993 | 10 | 63/37 | 91/9 | 0 | 7 |
| | Middle | 192 | 10 | 46/54 | 97/3 | 2 | 21 |
| | Distal | 69 | 8 | 54/46 | 92/8 | 20 | 9 |

*Unregistered treatment: 4%.

compared to 10% in women (p<0.001). 2% of all initially non-operated fractures in adults had secondary surgery due to failed primary treatment.

## Incidence

The incidence in the VG-region from 2015 to 2019 ranged from 6.6 to 9.3 per $10^4$ PYR (Table 3).

## PROMs

Complete responses at both the time of injury and after one year were 10% for EQ-5D and 14% for SMFA arm/hand score. Table 4 present change in PROMs scores according to finger, phalanx, sex, and treatment. There was no change in EQ-5d. Fractures of the 2nd finger had worse SMFA result than in the other fingers and operated patients reported worse SMFA than non-operatively treated patients. There were no differences between men and women regarding PROMs.

**Table 3. Incidence of finger fractures in Region Västra Götaland 2015–2019.**

| Year | No of registred fractures | Population | Incidence ($10^4$ PYR) |
|---|---|---|---|
| **2015** | 1096 | 1648682 | 6.6 |
| **2016** | 1267 | 1671783 | 7.6 |
| **2017** | 1280 | 1690782 | 7.6 |
| **2018** | 1274 | 1709814 | 7.5 |
| **2019** | 1611 | 1725881 | 9.3 |

**Table 4. Patient-reported outcome measures (PROMs) changes from baseline to 1 year after injury.**

| | Change EQ-5D median (IQR) | p-value* | Change SMFA median (IQR) | p-value* |
|---|---|---|---|---|
| **All fractures** | 0 (-0.2–0) | 0.000 | 3 (0–13) | 0.000 |
| **Index finger** | 0 (-0.2–0) | 0.000 | 6 (0–16) | 0.000 |
| **Middle finger** | 0 (-0.2–0) | 0.000 | 3 (0–12) | 0.000 |
| **Ring finger** | 0 (-0.2–0) | 0.000 | 3 (0–13) | 0.000 |
| **Little finger** | 0 (-0.2–0) | 0.000 | 3 (0–9) | 0.000 |
| | p = 0.17** | | p = 0.001** | |
| **Basal phalanx** | 0 (-0.2–0) | 0.000 | 3 (0–13) | 0.000 |
| **Middle phalanx** | 0 (-0.2–0) | 0.000 | 3 (0–13) | 0.000 |
| **Distal phalanx** | 0 (-0.2–0) | 0.000 | 3 (0–9) | 0.000 |
| | p = 0.273** | | p < 0.001** | |
| **Men** | 0 (-0.2–0) | 0.000 | 3 (0–9) | 0.000 |
| **Women** | 0 (-0.2–0) | 0.000 | 3 (0–13) | 0.000 |
| | p = 0.438*** | | p < 0.001*** | |
| **Non-operatively** | 0 (-0.2–0) | 0.000 | 3 (0–9) | 0.000 |
| **Operatively** | 0.04 (-0.2–0) | 0.000 | 6 (0–16) | 0.000 |
| | p = 0.02*** | | p < 0.001*** | |

*Wilcoxon signed rank

**Kruskal-Wallis

***Mann-Whitney U-test.

## Discussion

Finger fractures are common injuries that are treated by orthopedic—and hand surgeons as well as emergency doctors and general practitioners. Based on registry data of 21341 finger fractures it was found that the most common fracture is in the base phalanx of the 5th finger. The distribution and location of different finger fractures have previously not been presented in detail, based on a large population. Earlier studies that describe anatomical distribution down to each single phalanx are based on only 800–1000 patients but show a similar pattern [1,11,23].

This study reports a higher mean age (40 years) than previous studies. Feehan et al 2006 [2] reported a mean age of 31 years for finger and metacarpal fractures, and Court-Brown et al 2006 [1] found a mean age of 36 years for finger fractures. This difference might be due to more comprehensive data in this study, differences in the population or cultural differences regarding activities such as work and sports. In accordance with these studies, it was found that finger fractures mostly affect relatively young patients and do not follow the pattern of osteoporotic fractures. The mean age differs between men and women, but they fracture their fingers in the same period of life (15–50 years of age). The direct cause of finger fractures has formerly not been presented in large epidemiological studies [2,18,21]. This study concludes that fall and crush injuries were the most common causes.

Most finger fractures were treated non-operatively and women were treated non-operatively to a larger extent than men. Fractures that were operated to a greater extent (mid and proximal phalanx of the 2nd finger) were not more common in men which implies that differences in fracture location do not explain this difference between the sexes. The difference can probably be explained by greater presence of open fractures in men. There is no information in the SFR about fracture dislocations or instability that affect the decision to operate or not, hence it was not possible to analyze if differences in these factors also might contribute to the

difference between men and women. Women were slightly older than men at the time of fracture, and age may influence choice of treatment. The difference between men and women in treatment may also reflect a potential inequality in the health care.

There are no reports of the minimum clinically important difference (MCID) in SMFA Arm/hand index. MCID for SMFA in ankle injuries have been estimated to 7 points [24]. For EQ-5D, MCID is estimated to 0.1 [25]. The observed differences in SMFA and EQ-5D in this study from pre-injury to one year between fracture types, fractured finger, sex, and treatment did not reach MCID and are, in this study, not considered as clinically relevant even if the differences in SMFA were statistically significant due to the large sample. Based on the available data it is interpreted, that one year after injury, hand function measured with the SMFA hand/arm index do not seem to be affected by a finger fracture in general. Neither does the quality of life, measured by EQ-5D, seem to be diminished. However, SMFA hand/arm index might not be sensitive enough to demonstrate hand disability due to finger fractures. Quality of life is a broad measure and is not affected by this type of injury.

The incidence could only be studied in one specific region of Sweden. Due to low coverage in the SFR in general the first year, incidence before 2015 are not reported. During the observed period, the incidence increased and 2019 it was 9,3 per $10^4$ PYR. Feehan and Sheps, 2006 [2] report an incidence of 18 per $10^4$ PYR for phalangeal fractures in British Columbia, Canada (4 million inhabitants). A similar study from the United States by Karl et al [26], reports an incidence of 12.5 per $10^4$ PYR (87 million inhabitants). The divergent incidence rates might be explained by differences regarding age, sex, and activity patterns between the various populations. There is no indication of such difference between these three populations. The difference in incidence most likely reflects that not all fractures are yet registered in the SFR. The increasing incidence rates in this study are most likely due to successively improved coverage in the register. Incidence calculated from SFR data could potentially, when coverage is better, give a very accurate view, given that its data is based on whole heterogenous population of a country.

Children were not included in the SFR until 2015 which affected the analysis of age and incidence, where only data from 2015–2019 could be used. However, fracture type and treatment were analyzed for children and adults separately and the whole data set could then be used. Only 10% and 14% of the patients in the registry completed the one-year follow-up regarding PROMs. The large amount of missing data for PROMs questionnaires is a known challenge for quality registries. The question is whether the responses from the minority of patients that answers the questionnaires reflect the general population or if the results are biased due to missed responses. If responses are missed at random and not due to a systematic reason, the actual responses can be considered as representative for the whole sample. An analysis of 317 non-responders in the SFR by Juto et al [27] indicated that both in the preinjury survey as well as in the one-year survey, non-responders in the reported similar EQ-5D and SMFA scores compared to responders. This study suggests that the missing data in SFR is not caused by the fact that non-responders are more (or less) discontent. Despite a low PROMs response rate, the number of complete answers were substantial (2146 and 2992, respectively). Together with low response rates the main limitation of this study is that the SFR still does not have full national coverage which makes it difficult to estimate accurate incidence rates. Within recent years most orthopedic and trauma units in Sweden have affiliated to the SFR which means that future studies will be able to present more accurate incidence rates.

Based on the extensive registry data from the SFR this study presents detailed epidemiological information about finger fractures that can be used as a point of reference in clinical work and for future studies.

## Supporting information

**S1 Checklist. STROBE checklist.**
(PDF)

**S1 File. Complete data set.**
(SAV)

## Author Contributions

**Conceptualization:** Maria Wilcke.

**Data curation:** Henrik Alfort, Johanna Von Kieseritzky, Maria Wilcke.

**Formal analysis:** Henrik Alfort, Johanna Von Kieseritzky, Maria Wilcke.

**Funding acquisition:** Johanna Von Kieseritzky.

**Methodology:** Maria Wilcke.

**Supervision:** Maria Wilcke.

**Visualization:** Henrik Alfort.

**Writing – original draft:** Henrik Alfort.

**Writing – review & editing:** Johanna Von Kieseritzky, Maria Wilcke.

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
