## [Decision Letter · Decision Letter 0]

4 Apr 2023

PONE-D-23-02792Finger fractures: Epidemiology and treatment based on 21341 fractures from the Swedish Fracture RegisterPLOS ONE

Dear Dr. Alfort,

Thank you for submitting your manuscript to PLOS ONE. After careful consideration, we feel that it has merit but does not fully meet PLOS ONE’s publication criteria as it currently stands. Therefore, we invite you to submit a revised version of the manuscript that addresses the points raised during the review process.

We look forward to receiving your revised manuscript.

Kind regards,

Dario Piombino-Mascali, Ph.D.

Academic Editor

PLOS ONE

Additional Editor Comments:

Dear all, please address the concerns raised by the reviewers, and I will be happy to consider a revised version of this article. Please note that it would be appropriate to have the manuscript read by a native English speaker prior to submission.

Best wishes,

Reviewers' comments:

Reviewer's Responses to Questions

**Comments to the Author**

1. Is the manuscript technically sound, and do the data support the conclusions?

Reviewer #1: Yes

Reviewer #2: Partly

2. Has the statistical analysis been performed appropriately and rigorously? 

Reviewer #1: Yes

Reviewer #2: Yes

3. Have the authors made all data underlying the findings in their manuscript fully available?

Reviewer #1: Yes

Reviewer #2: Yes

4. Is the manuscript presented in an intelligible fashion and written in standard English?

Reviewer #1: Yes

Reviewer #2: Yes

5. Review Comments to the Author

Reviewer #1: PONE-D-23-02792 - Finger fractures: Epidemiology and treatment based on 21341 fractures from the Swedish Fracture Register

The submitted manuscript describes epidemiology and treatment of finger fractures from the Swedish Fracture Register (SFR) in the years 2015-2019. While it may potentially represent an interesting contribution to epidemiological studies on fractures as a specific medical condition, the main issue of this study in its current form is that there is no clear research questions/hypothesis. With regards to, how should this study contribute to the understanding of finger fractures more broadly? What is the importance of epidemiological fracture data? As a research article, it needs more contextualization.

Below, I offer a number of comments and questions about the meaning of some content.

Keywords. Please remove ‘finger fracture’. Keywords and title should not include the same words.

Introduction

Lines 58-60. The aim and objectives should be placed in the framework of a research paper that offer novel information to fill a knowledge gap. As it stands, the work is declared as descriptive and it is unclear how scholars may benefit from epidemiology of SFR. Why is important having epidemiological studies on finger fractures? Which is the problem the authors may want to contribute with their research? How epidemiology from SFR contribute to the understanding of finger fractures more broadly?

Some of these arguments are marginally reported in Lines 42-44.

Materials and Methods

Line 93. Please specify the program used for statistics.

Line 94. Is the term ‘gender’ used as synonym of ‘sex’?

Line 97. The statistical test reads ‘Kruskal-Wallis’. Please check throughout the manuscript.

Results

Line 103. Caption to Figure 1 should include more details.

Lines 107-108. ‘The age distribution according to gender is presented in figure 2.’ Please add a comment within the text.

Line 117. Table 1 ‘joint engagement’ odd word choice.

Line 141. Table 4. Please pay attention to issues of spacing (either side of the = sign)

Discussion

Line 160. Which are these ‘previous epidemiological studies’? Please add references.

Line 174. ‘ankle’ and not ‘ancle’.

Lines 187-191. Why are additional parameters (e.g. sex, age of the patients) not taken into account when comparing with other studies? More contextualization of SFR data is necessary. Finally, consideration of potential limits in comparability of results should be included.

Line 198. Juto et al. (2017) is not included within the final references.

Figure 1. 20337+1007 = 21334 and not 21341. Please revise.

Figure 2. Age range 0-99 years but Results section reports 0-101 years. Please revise.

Reviewer #2: The paper is well written and the data clearly presented.

It would be needed a better review of the state of the art in the introduction, whereas similar studies are only briefly mentioned in lines 186-189.

A better comparison between the current study and previous studies could also benefit and improve data readability for future studies.

6. PLOS authors have the option to publish the peer review history of their article (what does this mean?). If published, this will include your full peer review and any attached files.

Reviewer #1: No

Reviewer #2: No

---

## [Author Response · Author response to Decision Letter 0]

24 Apr 2023

As requested by the editor please see separately uploaded file named Response to reviewers.

---

## [Decision Letter · Decision Letter 1]

9 May 2023

PONE-D-23-02792R1Finger fractures: Epidemiology and treatment based on 21341 fractures from the Swedish Fracture RegisterPLOS ONE

Dear Dr. Alfort,

Thank you for submitting your manuscript to PLOS ONE. After careful consideration, we feel that it has merit but does not fully meet PLOS ONE’s publication criteria as it currently stands. Therefore, we invite you to submit a revised version of the manuscript that addresses the points raised during the review process.

We look forward to receiving your revised manuscript.

Kind regards,

Filippo Migliorini MD, PhD, MBA

Academic Editor

PLOS ONE

Reviewer #2: The Authors successfully incorporated the suggested revision. However, a few grammatical (e.g., ‘data’ takes the plural form of a verb or pronoun as ‘datum’ is the singular form) errors still persist in the revised manuscript.

Comments: Figure 2. As per my earlier comment, x-axis values and age range as reported in the main text should match, therefore I suggest the Authors to add the number 102 as an extra mark.

Academic Editor Notes: Dear authors, thank you for your contribution. There are some points which should be further addressed before formal acceptance:

1. upgrade your manuscript to the STROBE guidelines. Readapt carefully the subheadings. State the use of the STROBE guidelines and cite them. Attach the STROBE checklist as supplementary material

2. abbreviation should be clarified at once, then use only the mentioned abbreviation (e.g. PROMs)

3. Divide methods and results into subheadings.

4. PROMs not PROM!

5. PROMs not PROM scores!

6. when you give percentages, you need also to clarify the number of events/observations. For example 20% (20 of 100)

7. Use ALWAYS third person and passive voice

8. ABSTRACT:

8.1. Add the conclusion!

9. METHODS:

9.1. Declare in DETAIL that you follow the principles expressed in the Helsinki declarations AND later amendments, the signed consent of patients must be declared.

9.2. How you evaluated the fracture classification? This is one of the most important information that must be explained IN DETAIL, and potential limitations must also be acknowledged.

9.3. Report a figure with the classification system you used. Cite the classification system.

9.4. Report the therapeutic framework in DETAIL and add a figure of it in the methods

10. RESULTS:

10.1. Describe in DETAIL the identification process, with the exact excluded and included patients. Add these also to the flowchart.

10.2. If you can add some Figures describing your results will be appreciated. These results are a lot of numbers and figures that could help to summarise your findings

11. DISCUSSIONS:

11.1. Limitations have not been identified. Please create a paragraph of 250-500 words identifying all possible limitations

12. CONCLUSIONS:

12.1. There are no conclusions in support of your findings. Please elaborate a strong evidence-based conclusion

---

## [Author Response · Author response to Decision Letter 1]

14 Jun 2023

Letter with respone so all questions are uploaded as described in the decision letter

---

## [Editor Report · Decision Letter 2]

16 Jun 2023

PONE-D-23-02792R2Finger fractures: Epidemiology and treatment based on 21341 fractures from the Swedish Fracture RegisterPLOS ONE

Dear Dr. Alfort,

Thank you for submitting your manuscript to PLOS ONE. After careful consideration, we feel that it has merit but does not fully meet PLOS ONE’s publication criteria as it currently stands. Therefore, we invite you to submit a revised version of the manuscript that addresses the points raised during the review process.

We look forward to receiving your revised manuscript.

Kind regards,

Filippo Migliorini

Academic Editor

PLOS ONE

Journal Requirements:

**Additional Editor Comments:**

Dear Authors,

Just want to ask you for some minimal revisions:

Remove the subheading "objectives" in the introduction section

divide the methods section into subheadings: Study protocol (add all your declarations: Helsinki, STROBE, ethics etc), Eligibility criteria, Outcome assessment, Statistical analysis

divide also the Results into subheadings according to your findings

Remove subheadings in the discussion section

Limitations: discuss the lack of children until 2015 and its possible effect

Improve the scientific language level and use the third person. There are still several points which could be improved

Thank you,

Filippo Migliorini

Editor

---

## [Author Response · Author response to Decision Letter 2]

26 Jun 2023

A letter with responses to the reviewers has been uploaded

---

## [Editor Report · Decision Letter 3]

28 Jun 2023

Finger fractures: Epidemiology and treatment based on 21341 fractures from the Swedish Fracture Register

PONE-D-23-02792R3

Dear Dr. Alfort,

We’re pleased to inform you that your manuscript has been judged scientifically suitable for publication and will be formally accepted for publication once it meets all outstanding technical requirements.

Kind regards,

Filippo Migliorini MD, PhD, MBA

Academic Editor

PLOS ONE

Additional Editor Comments (optional):

well done
---

## [Editor Report · Acceptance letter]

6 Jul 2023

PONE-D-23-02792R3 

Finger fractures: Epidemiology and treatment based on 21341 fractures from the Swedish Fracture Register 

Dear Dr. Alfort:

I'm pleased to inform you that your manuscript has been deemed suitable for publication in PLOS ONE. Congratulations! Your manuscript is now with our production department. 

Kind regards, 

on behalf of

Dr Filippo Migliorini 

Academic Editor

PLOS ONE